

# Ultrasonographic modeling of lung and diaphragm mechanics: clinical trial of a novel non-invasive method to evaluate pre-operative pulmonary function

Tianyuan Li[1], Xiong-zhi Wu[2], Dingde Long[1], Huan Fu[1], Suping Guo[3] and Fen Liu[4]

[1] Department of Anesthesiology, The First Affiliated Hospital, Jiangxi Medical College, Nanchang University, Nanchang, Jiangxi, China
[2] Department of Anesthesiology, Shaoxing People's Hospital, Shaoxing, Zhejiang, China
[3] Department of Ultrasound Medicine, The First Affiliated Hospital, Jiangxi Medical College, Nanchang University, Nanchang, Jiangxi, China
[4] Department of Intensive Care Unit, The First Affiliated Hospital, Jiangxi Medical College, Nanchang University, Nanchang, Jiangxi, China

Corresponding author
Fen Liu, liufenzzyx@163.com

## ABSTRACT

**Background:** Pre-operative pulmonary function testing (PFT) plays a key role in predicting postoperative complications or functional impairment. However, PFT requires the subject and examiner to cooperate and the results are influenced by both technical and personal factors. In contrast, the use of ultrasound (US) for structural and functional assessments of the lungs and diaphragm is on the rise, as it requires minimal patient cooperation. Dyspnea is mainly caused by lung or pleural lesions but may also be caused by weak respiratory muscles. As the diaphragm is a primary respiratory muscle, combining lung ultrasonography (LUS) with diaphragm ultrasound (DUS) may enable a more comprehensive assessement of pulmonary function. This study aims to introduce a novel approach for assessing pulmonary function using a mathematical model based on LUS and DUS.

**Methods:** This prospective study was performed at the First Affiliated Hospital of Nanchang University between June 2021 and December 2021, 208 patients were recruited and underwent PFT, LUS, and DUS examinations. An experienced physician, blinded to the clinical history and PFT results, performed LUS and DUS and explored the correlations between a mathematical model (ultrasonographic modeling score (U-score)) using LUS combined with DUS and pulmonary function parameters. Univariate, multivariate, and logistic regression analyses were also performed.

**Results:** According to the univariate and multivariable analysis, diaphragm thickness fraction in deep breathing (D-DTF) (odds ratio (OR), 0.88; 95% confidence interval (CI) [0.83–0.94]; $P < 0.001$), and LUS score (OR, 1.44; 95% CI [1.16–1.80]; $P < 0.001$) were each independently associated with pulmonary function. According to the logistics equation, a U-score of $-0.126 \times$ D-DTF $+ 0.368 \times$ LUS score was produced. The U-score showed a more significant negative correlation with forced expiratory volume in the first second/forced vital capacity (FEV1/FVC) ($r = -0.605$, $P < 0.001$) than the LUS or DUS indices alone. The U-score (area under the curve (AUC) = 0.971) was greater than the other indices for assessing pulmonary function.

**Conclusions:** With validation, the U-score through both lung and diaphragm ultrasound measurements may assist in estimating pulmonary function. This approach facilitates the assessment of pulmonary function in patients who may be unable to reliably participate in PFT.

## INTRODUCTION

Pulmonary function testing (PFT) is an important part of surgical evaluation of patients, and helps to monitor impaired function (*Ruppel & Enright, 2012*). Pre-operative PFT parameters including forced expired volume in the first second as a fraction of forced vital capacity are important predictors of post-operative complications (*Matsumi et al., 2017*; *Zhang et al., 2015*). However, the efficacy of PFT is constrained by the patient's ability to comprehend the test's fundamentals and to cooperate with the examiner (*Tojo, 2006*). Therefore, there is a pressing need to identify alternative methods for assessing pulmonary function that can mitigate these limitations. The aim of this study was to develop objective criteria to identify patients at high risk for pulmonary dysfunction.

Ultrasound (US) is a viable and non-invasive assessment instrument that only requires minimal cooperation from a patient, making it a promising tool for clinical applications.

Lung ultrasonography (LUS) is a valuable diagnostic tool that complements physical examinations across a variety of pulmonary disease states and clinical settings (*Volpicelli et al., 2012*). Its capability to detect alterations in both the lung parenchyma and pleural cavities enhances its accessibility. Numerous studies have demonstrated that LUS offers significant benefits in the diagnosis of various respiratory diseases, including acute respiratory distress syndrome (*Stefanidis et al., 2011*), pulmonary edema (*Gargani et al., 2008*), interstitial lung disease (*Pitsidianakis et al., 2022*), pneumothorax (*Laursen et al., 2021*), atelectasis (*Monastesse et al., 2017*), pneumonia (*Ma et al., 2023*), and others. Furthermore, LUS proves beneficial for diagnosing and assessing lung diseases in specific populations, such as newborns (*Ma et al., 2023*), children (*Amatya et al., 2023*), and pregnant women (*Piccolo et al., 2023*). While LUS can serve as a key diagnostic and evaluation tool for a range of respiratory conditions, it is important to note that the comprehensive evaluation of pulmonary function necessitates both lung and respiratory muscle assessments. Consequently, LUS alone is insufficient for accurately evaluating pulmonary function.

LUS examines the lung fields and pleura, but does not assess respiratory muscle function. Diaphragm ultrasound (DUS) is widely utilized to assess diaphragmatic activity. Several studies have indicated a correlation between the ultrasound indices of the diaphragm and lung function parameters across various conditions, including osteoporosis, vertebral fractures, kyphosis, neuromuscular disease (*DePalo & McCool, 2002*), patients with stroke (*Chen et al., 2023*), chronic obstructive pulmonary disease (COPD) (*Topcuoğlu et al., 2022*), and healthy subjects (*Cardenas et al., 2018*). US

measurements of diaphragm excursion (DE) and diaphragm thickening fraction (DTF) have been validated as effective means for assessing diaphragmatic function (*Tralhão et al., 2020*; *Unal et al., 2000*).

In recent decades, LUS and DUS have been utilized to evaluate the characteristics of the lung and diaphragm, respectively, and serve as diagnostic methods for thoracic diseases that require minimal patient cooperation. While neither LUS nor DUS can fully assess pulmonary function independently, the potential for their combination to quantify pulmonary function remains unexplored. The primary objective of our prospective cross-sectional study was to develop a novel approach for assessing pulmonary function using a mathematical model based on LUS combined with DUS patterns in a cohort of perioperative patients by exploring the correlations between the ultrasound indices and PFT, thereby providing a non-invasive alternative for evaluating lung function in perioperative patients who may not fully comprehend or cooperate with the examiner.

## MATERIALS AND METHODS

### Participants

This prospective cross-sectional study was approved by the Ethics Committee of the First Affiliated Hospital of Nanchang University (No. 2021-8-001). All participants provided written informed consent before participation. The trial was registered at chictr.org.cn before enrolment (No: ChiCTR2100048032; Principal Investigator: T.Y.L; registration date: June 28, 2021). This prospective cross-sectional study was performed at the First Affiliated Hospital of Nanchang University, China, from 6/2021, 12/2021. Patients aged >18 years who underwent elective surgery and consented to undergo PFT, LUS, and DUS were included. The exclusion criteria included patients not providing consent, emergency surgery, body mass index exceeding 35 kg/m$^2$, history of abdominal and/or thoracic surgery, and thoracic deformities or scoliosis.

### Study procedures

Patients underwent PFT 1 day prior to the surgical procedure and a detailed record of the PFT parameters was made including FVC, FEV1/FVC, maximum voluntary ventilation (MVV), maximum expiratory flow at 50% of lung capacity (MEF50), residual capacity/total lung capacity (RVTLC), diffusion capacity for carbon monoxide of the lung (DLCO). On the same day, blinded to the pulmonary function test results, all clinical data and previous readings, the patients underwent lung and diaphragm ultrasound examinations conducted by two ultrasound specialists with extensive experience in accordance with the following methodology, with an interval of 5 h between US examinations and PFT. The ultrasound variables including lung ultrasound score (LUSs), quiet breathing diaphragmatic thickening during expiration (Q-DTe), quiet breathing diaphragmatic thickening during inhalation (Q-DTi), deep breathing diaphragmatic thickening during expiration (D-DTe), deep breathing diaphragmatic thickening during inhalation (D-DTi), quiet breathing diaphragmatic excursion (Q-DE), deep breathing diaphragmatic excursion (D-DE), quiet breathing diaphragmatic thickening fraction (Q-DTF), and deep breathing diaphragmatic thickening fraction (D-DTF). Subsequently, the correlations between the

ultrasound values and PFT parameters were explored to develop a novel non-invasive method for evaluating pulmonary characteristics, utilizing a mathematical model based on LUS combined with DUS patterns. This mathematical model was constructed using the logistic equation (U-score = $\Sigma\beta i\ Xi$; where $\beta i$ represents the coefficient of the variable Xi, and Xi denotes the independent risk factor) (*Nashef et al., 2012*).

## Pulmonary function test

All patients underwent standard pulmonary function testing using a MasterScreen diagnostic spirometer. FVC, FEV1/FVC, MVV, MEF50, RVTLC and DLCO were measured. According to the statement of the European Respiratory Society/American Thoracic Society (ERS/ATS) (*Brusasco, Crapo & Viegi, 2005*), FEV1/FVC < 70% is the criterion for irreversible lung function impairment, whereas >70% is considered normal pulmonary function.

## Lung ultrasonography

Patients underwent an LUS examination in accordance with a 12-zone protocol (*Soummer et al., 2012*), which was performed using a wisonic ultrasound machine (Wisonic Medical Co., Ltd., Shenzhen, China) equipped with a 2–5 MHz curvilinear probe in transversal scan by trained sonographers (T.Y.L., X.Z.W, and D.D.L.); intercostal space scanning was performed in each quadrant, and the corresponding ultrasonographic images were stored for further analysis. Twelve quadrants (six quadrants on the left and right lungs, respectively) were assessed by ultrasonography, as described previously (*Wu et al., 2022*). Each hemithorax was divided into anterior, lateral, and posterior areas that were separated by the anterior and posterior axillary lines, and divided in upper and lower portions that were separated by the boundary of 1 cm above the nipple. We evaluated lung aeration by using the LUS that was previously described (*Wu et al., 2022*) that based on the following scoring criteria (Fig. S1): ① Normal aeration (N): lung sliding sign and A lines or less than three isolated B line(s), marked as N; ② Moderate aeration loss (B1): multiple, vertical, laser-like B-lines or one or more small subpleural consolidations, marked as B1; ③ Severe aeration loss (B2): multiple merged B lines occupying the whole lung image (so-called "white lung") or multiple small subpleural consolidations, marked as B2; ④ Complete aeration loss (C): localized consolidation (subpleural tissue-like pattern), marked as C. A LUS for each quadrant was assigned as follows: $N = 0$, $B1 = 1$, $B2 = 2$ and $C = 3$. The scores of the 12 quadrants were added to calculate the total LUSs.

## Diaphragmatic ultrasonography

Diaphragm ultrasound has received more attention to assess respiratory function, as the diaphragm is the main respiratory muscle contributing more than 60% of the tidal volume in each breath (*Gargani et al., 2008*). Diaphragmatic exercise (DE) is the magnitude of downward and upward movement of the diaphragm during respiration, and a decrease in the magnitude of diaphragmatic exercise may indicate impaired diaphragmatic function. The diaphragm thickening fraction (DTF) is the percentage change in diaphragm thickness during inspiration and is an indicator of diaphragm contractility, which can help

to determine the degree of fatigue and functional status of respiratory muscles. DE and DTF have potential efficacy for diagnosing diaphragm dysfunction (*Wiesmann et al., 2016*; *Dubé & Dres, 2016*). A semi-recumbent position of 30° was used for all the participants. The diaphragm was scanned using a wisonic ultrasound machine (Wisonic Medical Co., Ltd., Shenzhen, China) between the anterior axillary line and midclavicular line in the right subcostal region. The left diaphragm was not assessed due to difficulty with image acquisition related to the gastric bubble and intestinal gas (Fig. S2) (*Dhungana et al., 2017*; *Vivier et al., 2012*; *DiNino et al., 2014*). Diaphragmatic thickness (DT) was measured in B-mode with a 12 MHz linear array probe over the zone of opposition the diaphragm from the pleural line to the peritoneal line during deep breathing (deep breathing-DT; D-DT) and quiet breathing (quiet breathing-DT; Q-DT). To determine diaphragm function, the DTF was calculated by measuring the ability of the diaphragm to contract according to the following formula: DTF = (DT at the end of inspiration–DT at the end of expiration)/DT at the end of expiration × 100%. DE was measured by a 2–5 MHz curvilinear probe placed along the midclavicular line or below the right costal margin during both deep breathing (deep-diaphragmatic excursion; D-DE) and quiet breathing (quiet-diaphragmatic excursion; Q-DE) with the sampling line and diaphragm as vertically as possible (at least 70° (*El-Halaby et al., 2016*)). DE (amplitude in cm, velocity in cm/s especially during sniffing) can be measured using M-mode ultrasonography. Subsequently, position markers were placed on the skin surface, three measurements were obtained, and the average was recorded.

## Statistical analysis

Statistical analyses were performed using SPSS version 23 software (SPSS Inc., Chicago, Illinois, USA). Continuous data were expressed as the mean ± SD or median (IQR). Continuous variables were analyzed using ANOVA or the Kruskal-Wallis test. Categorical data were expressed as frequencies or percentages. Chi-square or Fisher's exact test was used for comparison, as appropriate. Univariate logistic regression analysis was performed to identify the risk factors for abnormal pulmonary function. All variables associated with abnormal lung function, with $P$-value < 0.1 were candidates for backward stepwise multivariable analysis to identify independent risk factors. To predict abnormal lung function, a mathematical model was constructed using logistic regression algorithms to screen risk factors related to abnormal pulmonary function. According to the Hosmer-Lemeshow goodness of fit test, $P < 0.05$ is considered to be statistically significant and the degree of fit is good. The performance of the prediction model was evaluated in terms of sensitivity, specificity, accuracy, and area under the curve (AUC); receiver operating characteristic (ROC) analyses were performed to determine optimal cut-off values to detect the outcome. $P < 0.05$ was considered statistically significant.

## RESULTS

Of all 208 patients included in the study, 35 patients underwent thoracic surgery, 58 patients underwent urologic surgery, 65 patients underwent laparoscopic surgery, and 50 patients underwent breast surgery. Among these, 174 patients exhibited normal

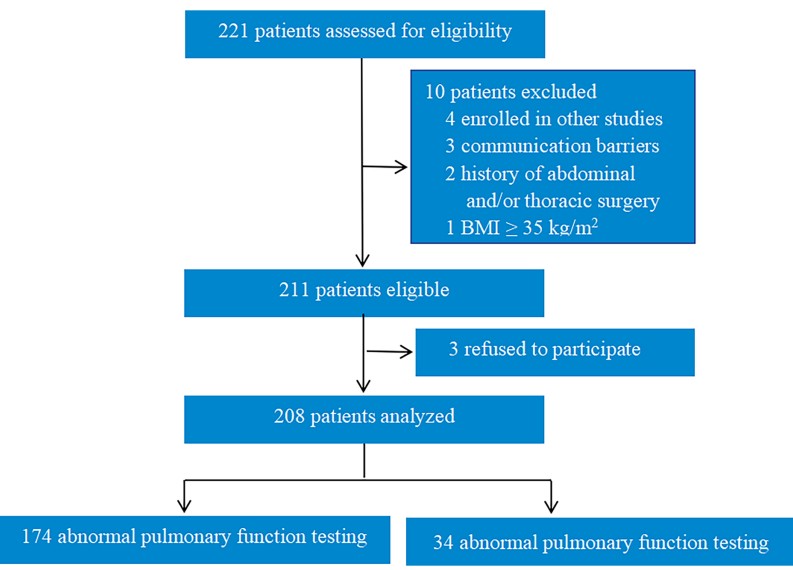

**Figure 1** Trial diagram.               

**Table 1 Baseline characteristics of all participants.**

| Variable | Normal (N = 174) | Abnormal (N = 34) | F/χ² | P value |
|---|---|---|---|---|
| Age (yr) | 62.25 ± 11.38 | 67.82 ± 8.84 | 7.28 | 0.008 |
| Gender, n (%) | | | 0.19 | 0.798 |
| Male | 90 (51.72%) | 19 (55.88%) | | |
| Female | 84 (48.18%) | 15 (44.12%) | | |
| BMI (Kg/m²) | 23.29 ± 3.45 | 22.71 ± 3.08 | 0.83 | 0.362 |
| Q-DE (cm) | 2.15 ± 0.21 | 1.91 ± 0.44 | 23.78 | 0.001 |
| D-DE (cm) | 5.20 ± 0.47 | 4.01 ± 1.05 | 113.70 | 0.001 |
| Q-DTF (%) | 41.22 ± 10.23 | 26.41 ± 7.07 | 65.13 | 0.001 |
| D-DTF (%) | 72.84 ± 16.36 | 41.08 ± 7.39 | 122.60 | 0.001 |
| LUS | 1.85 ± 2.10 | 7.06 ± 2.79 | 166.24 | 0.001 |

Note:
   BMI, Body mass index; Q-DE, Quiet breathing diaphragmatic excursion; D-DE, Deep breathing diaphragmatic excursion; Q-DTF, Quiet breathing diaphragmatic thickening fraction; D-DTF, Deep breathing diaphragmatic thickening fraction; LUSs, Lung ultrasound score.

pulmonary function testing, while 34 patients had abnormal pulmonary function (Fig. 1). As shown in Table 1, the abnormal pulmonary function group had a greater mean age (67.82 ± 8.89 *vs*. 62.25 ± 11.38, *P* < 0.001), a higher LUS score (7.06 ± 2.79 *vs*. 1.85 ± 2.10, *P* < 0.001), a lower DE (Q-DE:1.91 ± 0.44 *vs*. 2.15 ± 0.21; D-DE: 4.01 ± 1.05 *vs*. 5.20 ± 0.47; *P* < 0.001), and a lower DFT (Q-DFT: 26.41 ± 7.07 *vs*. 41.22 ± 10.23; D-DFT: 41.08 ± 7.39 *vs*. 72.84 ± 16.36; *P* < 0.001) compared to the normal pulmonary function group. There were no significant differences in sex or body mass index between the two groups.

The analysis of the relationships among the variables revealed a strong negative correlation between the LUS score and several pulmonary function parameters, including FEV1/FVC, maximum expiratory flow at 50% of lung capacity (MEF50), FVC, maximum

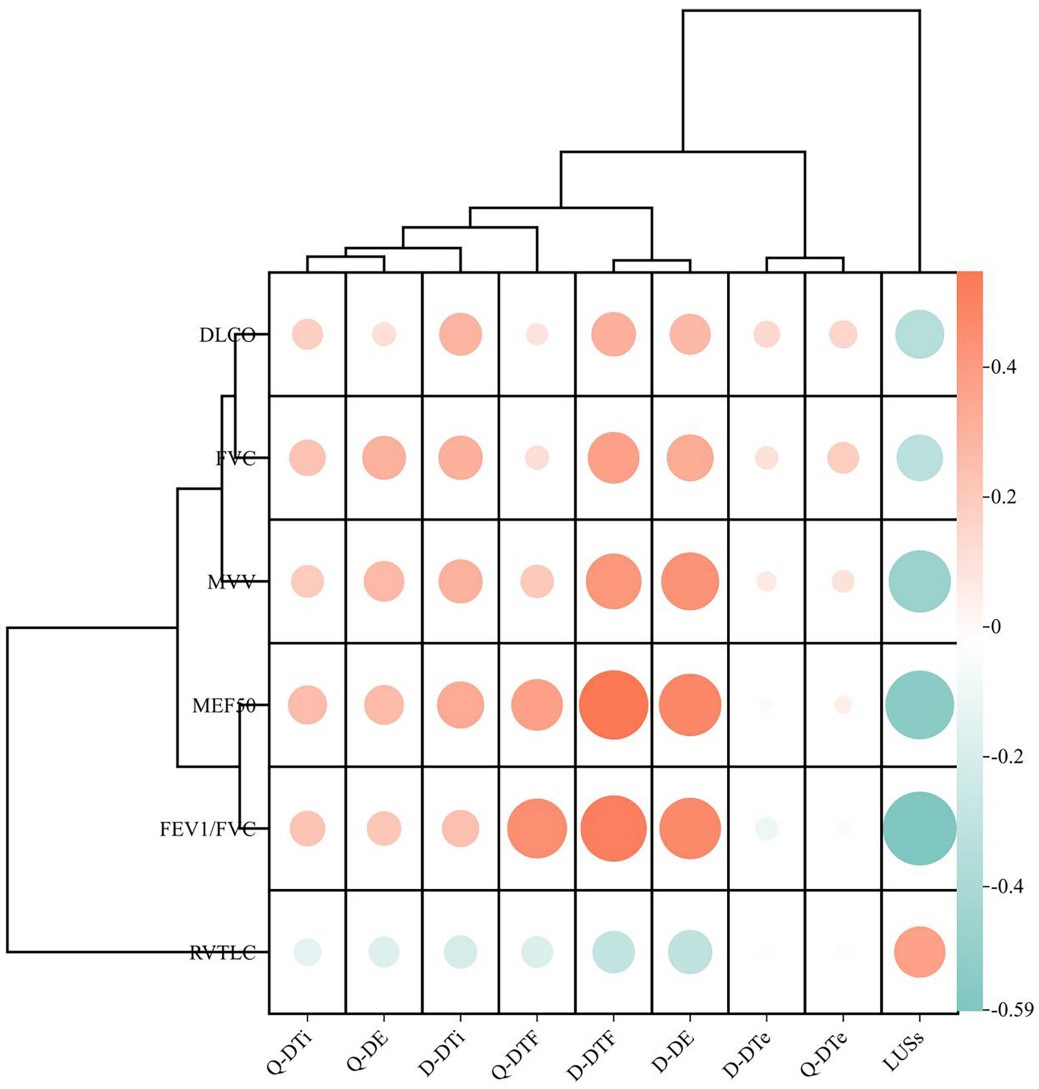

**Figure 2  Correlation between the ultrasound variables and pulmonary function parameters.** DLCO, Diffusion Capacity for Carbon Monoxide of the Lung; FVC, Forced vital capacity; FEV, Forced expiratory volume; MVV, Maximal ventilation volume; MEF, Maximum expiratory flow; RVTLC, Residual capacity/Total lung capacity; Q-DTe, Quiet breathing diaphragmatic thickening during expiration; Q-DTi, Quiet breathing diaphragmatic thickening during inhalation; D-DTe, Deep breathing diaphragmatic thickening during expiration; D-DTi, Deep breathing diaphragmatic thickening during inhalation; Q-DE, Quiet breathing diaphragmatic excursion; D-DE, Deep breathing diaphragmatic excursion; Q-DTF, Quiet breathing diaphragmatic thickening fraction; D-DTF, Deep breathing Diaphragmatic thickening fraction; LUSs, Lung ultrasound score.

voluntary ventilation (MVV), and DLCO ($P < 0.001$). Conversely, D-DE, Q-DTF, and D-DTF exhibited significant positive correlations with FEV1/FVC, MEF50, FVC, and MVV (Fig. 2). Notably, no other US variables correlated with the pulmonary function parameters. In the univariate logistic regression analysis, six variables were strongly associated with the risk of pulmonary dysfunction: age, Q-DE, D-DE, Q-DTF, D-DTF, and LUS score. The multivariable analysis indicated that D-DTF (odds ratio (OR), 0.88; 95% confidence interval (CI), [0.83–0.94]; $P < 0.001$) and LUS score (OR, 1.44; 95% CI,

**Table 2 The risk factors of pulmonary dysfunction in univariate and multivariate analysis.**

| Variable | Univariate analysis | | | Multivariate analysis | | |
|---|---|---|---|---|---|---|
| | OR | 95% CI | *P* value | OR | 95% CI | *P* value |
| Age | 1.06 | [1.01–1.10] | 0.019* | | | |
| Q-DE | 0.03 | [0.01–0.15] | 0.001** | | | |
| D-DE | 0.06 | [0.03–0.15] | 0.001** | | | |
| Q-DTF | 0.86 | [0.81–0.90] | 0.001** | | | |
| D-DTF | 0.85 | [0.80–0.89] | 0.001** | 0.88 | [0.83–0.94] | 0.001** |
| LUS | 1.95 | [1.60–2.38] | 0.001** | 1.44 | [1.16–1.80] | 0.001** |

Notes:
CI, confidence interval; OR, odds ratio.
* $P < 0.05$.
** $P < 0.01$.

[1.16–1.80]; $P < 0.001$) were independently associated with pulmonary function in our study (Table 2).

Based on the transformation of independent risk factors and the logistics equation as described by *Nashef et al. (2012)*, we developed an ultrasonographic modeling score (U-score) represented by the equation U-score = −0.126 × D-DTF + 0.368 × LUS score. The model was evaluated using the Hosmer-Lemeshow goodness-of-fit test, which yielded a result of $P > 0.05$, indicating that the model passed the test.

Furthermore, the U-score demonstrated a more significant negative correlation with FEV1/FVC ($r = −0.605$, $P < 0.001$) compared to the LUS or DUS index when assessed individually (Fig. 3). The AUC of the U-score was greater than that of the other indices in the assessment of patients' pulmonary function (Fig. 4, Table 3).

Ultimately, two sets of outliers were identified in the course of our trial. Two patients exhibited normal PFT but LUS scores of 9 and 17, respectively, and D-DTF values of 63.64% and 77.27%. The U-scores for these two patients were −4.71 and −3.48, respectively, which were greater than the cut-off value (−6.086).

# DISCUSSION

Pre-operative PFT are essential assessment tools for predicting pulmonary complications (*Wu et al., 2021*; *Matsumi et al., 2017*; *Liu et al., 2019*; *van Huisstede et al., 2013*). An abnormal PFT value, specifically a FEV1/FVC ratio of less than 70%, remains the sole predictive covariable for postoperative complications (*Yoshimi et al., 2016*). Among elderly patients with increasingly complex health conditions, reduced mobility, inadequate communication skills, cognitive impairment, ventilator dependency, or poor cooperation may affect the ability to complete PFT. Although pre-operative PFT is important for predicting postoperative complications, but the effectiveness is low in the elderly patients described above (*Tojo, 2006*). These patients need an alternative method to assess pulmonary function. The common non-invasive alternative to pulmonary function testing is the breath-holding test (*Sylvester et al., 2020*), but the result of the breath-holding test can be influenced by cardiorespiratory abnormalities (*Sharma & Shekh, 2017*) and should be judged on a clinical basis. It is important to note that some patients, despite some degree

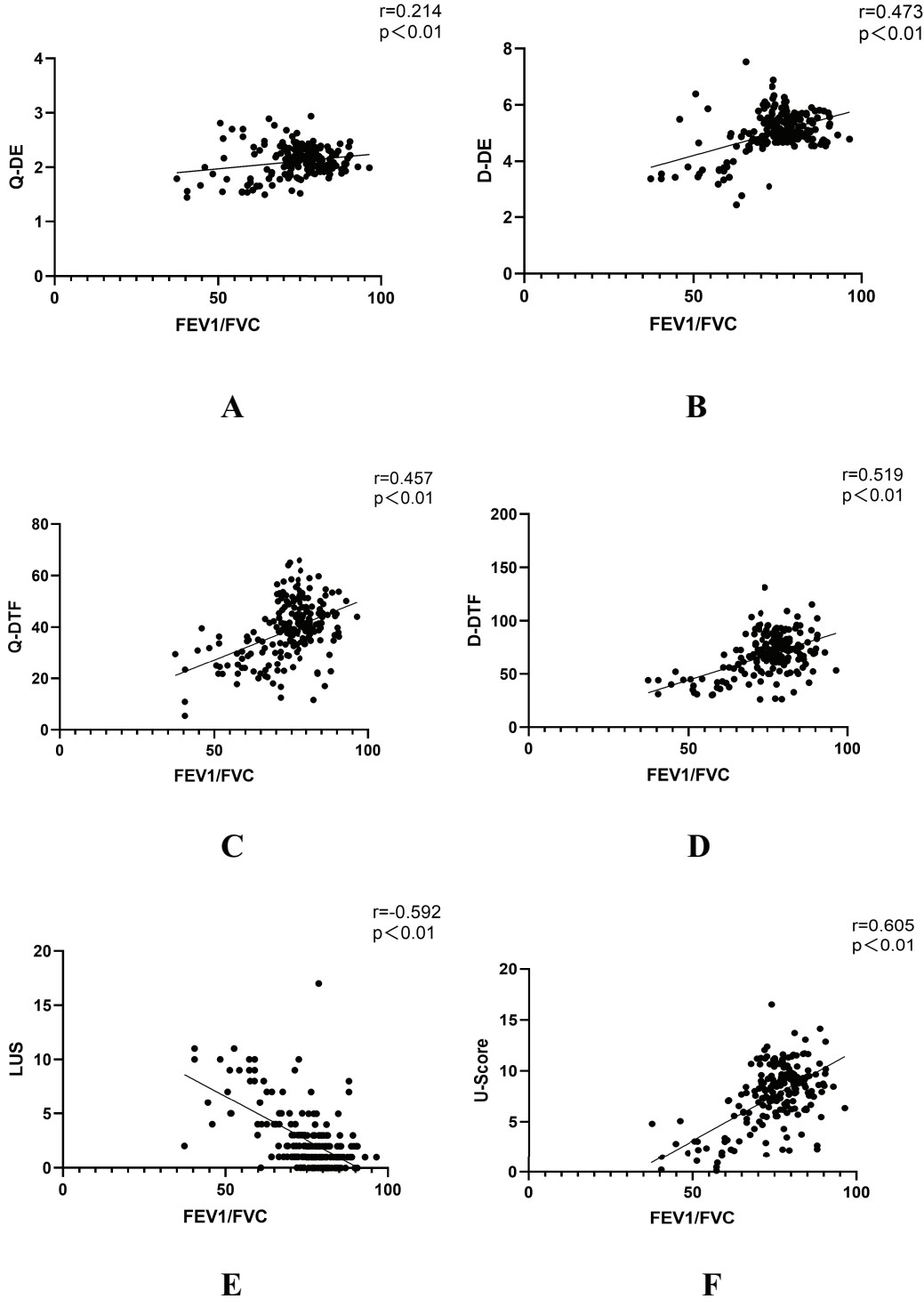

**Figure 3   Correlation between ultrasound parameters and FEV1/FVC.** (A) Correlation between quiet breathing diaphragmatic excursion (Q-DE) and FEV1/FVC. (B) Correlation between deep breathing diaphragmatic excursion (D-DE) and FEV1/FVC. (C) Correlation between quiet breathing diaphragmatic thickening fraction (Q-DTF) and FEV1/FVC. (D) Correlation between deep breathing diaphragmatic thickening fraction (D-DTF) and FEV1/FVC. (E) Correlation between lung ultrasound score (LUS) and FEV1/FVC (F) Correlation between ultrasonographic modeling score (U-score) and FEV1/FVC.

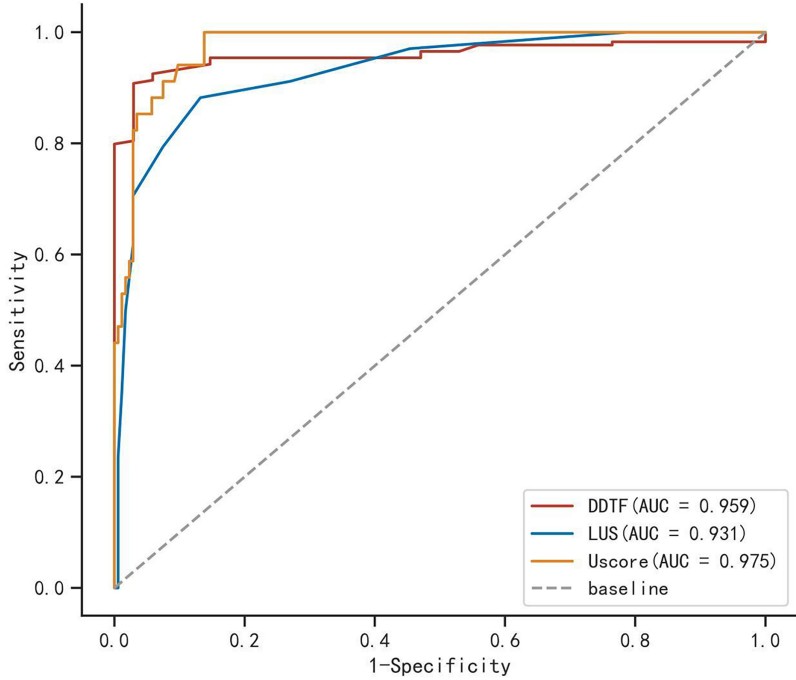

**Figure 4 Receiver-operating characteristic curve of D-DTF, LUS and U-score.** D-DTF, Deep breathing diaphragmatic thickening fraction; LUS, Lung ultrasound score, U-score, Ultrasonographic modeling score.

**Table 3 ROC curve analysis of D-DTF, LUS and U-score to identify abnormal lung function.**

| Variable | Cutoff | AUC | 95% CI | Sensitivity | Specificity |
|---|---|---|---|---|---|
| D-DFT | 52.087 | 0.959 | [0.933–0.985] | 0.971 | 0.908 |
| LUSs | 3.500 | 0.931 | [0.883–0.979] | 0.882 | 0.868 |
| U-score | −6.086 | 0.975 | [0.958–0.993] | 1.000 | 0.862 |

**Note:**
D-DTF, Deep breathing diaphragmatic thickening fraction, LUSs, Lung ultrasound score, U-score, Ultrasonographic modeling score.

of abnormality on routine pulmonary function tests, may have the duration of the breath-holding within the normal range due to training in breath-holding (*e.g.*, buteyko breathing technique (*Mavkar & Shukla, 2024*)), which may not be consistent with the PFT. Our trial developed a novel non-invasive approach for assessing pulmonary function using a mathematical model based on LUS combined with DUS patterns which only requires a minimal cooperation from patients.

Ultrasound is a viable and non-invasive assessment instrument that is independent of the features controlled by the patient and can be used as an alternative method to assess lung function and overcome these limitations. In this study, we have devised statistical models using US measurement to predict pulmonary function during surgery and identify major risk factors, providing an additional objective criterion for identifying patients at high risk of pulmonary dysfunction. Furthermore, according to the univariate and multivariable analysis, D-DTF (OR, 0.88; 95% CI [0.83–0.94]; $P < 0.001$), and LUS score

(OR, 1.44; 95% CI [1.16–1.80]; *P* < 0.001) were each independently associated with pulmonary function in our study. We used the independent risk factors to produce a mathematical model that estimates pulmonary function. The variables incorporated into this model can be easily obtained in clinical settings, facilitating their extension and application.

In accordance with previous studies (*Cardenas et al., 2018*; *Cohen et al., 1994*), we found positive correlations between DE and lung volume (FVC and FEV1), but sex influenced these correlations. DE more significantly correlated with female sex. Beyond the known differences in airway structure between sexes (*Harms, 2006*), it has already been reported that females also have smaller dimensions of the rib cage and, during QB, they have lower tidal volume, minute ventilation and abdominal contribution than males (*Romei et al., 2010*). In this study, DE was not an independent risk factor in multivariate regression, this can be attributed to the no difference in sex between the two groups. But it may be an area that requires further study.

Our study revealed a noteworthy correlation between FEV1/FVC (one of the most important variables of PFT) and D-DTF (r = 0.519, *P* < 0.001), LUS score (r = −0.592, *P* < 0.001). A similar correlation has been observed in a previous study (*Cardenas et al., 2018*; *Zheng, Wu & Tan, 2016*). The correlation between D-DTF, LUS and FEV1/FVC suggests that ultrasound measurements can provide an accurate pulmonary function assessment in patients who cannot cooperate with PFT. This is especially valuable for patients who cannot go anywhere or are confused. One of the key findings of the present study was that U-score (r = −0.605, *P* < 0.001) has a stronger correlation with FEV1/FVC than D-DTF and LUS, respectively. ROC curve analysis showed that both D-DTF and LUS score were accurate in predicting abnormal lung function results; however, the U-score had a larger AUC (0.975) and higher specificity and sensitivity for predicting pulmonary dysfunction when compared to both D-DTF and LUS score. This suggests that the combination of LUS and DUS offers enhanced predictability in preoperative lung function assessments. Multiple studies have confirmed that LUS is highly accurate for diagnosing many lung diseases (*Stefanidis et al., 2011*; *Gargani et al., 2008*; *Pitsidianakis et al., 2022*; *Laursen et al., 2021*; *Monastesse et al., 2017*), and DUS is a novel technique for assessing diaphragmatic movement (*Cardenas et al., 2018*; *Haji et al., 2018*; *Boussuges, Gole & Blanc, 2009*). However, the current investigation demonstrated that the use of LUS or DUS alone is limited in predicting preoperative lung function. The U-score combines LUS and DUS results to obtain a comprehensive assessment of lung function with high specificity and sensitivity for pulmonary dysfunction. This overcomes one of the limitations of these tools. Two patients had normal PFT but with LUS score >9 in the present study. All the patients had a history of bronchiectasis or infection. Their DUS parameters were normal, whereas LUS showed multiple solid signs (bronchial inflation, fragmentation, and tissue-like signs) in the lesion. The U-score of all two patients were greater than the cut-off value (−6.086), indicating that two patients had abnormal lung function and that they had a significantly higher probability of developing

postoperative pulmonary complications. During the post-operative follow-up, they developed varying degrees of post-operative lung infection, resulting in prolonged hospitalization in line with our predictions. A potential explanation for these findings is that PFT assess the overall function of both lungs and are not capable of evaluating the functional indices of individual lung lobes. Furthermore, PFT may exhibit a time lag, as they typically only become abnormal when the lung tissue has sustained significant damage (*Akira et al., 2009*).

This model can be used to assess the pulmonary dysfunction in patients undergoing perioperative surgery. The higher the total number of points in the model, the greater is the likelihood of pulmonary dysfunction. The model's indicators are conveniently repeatable in clinical settings, facilitating the evaluation of whether patients are susceptible to pulmonary dysfunction, enabling the early identification of at-risk patients, as well as early detection, intervention, and improvement of post-operative outcomes.

The present study had some limitations. First, LUS and DUS are operator-dependent methods, it is necessary to improve the accuracy of ultrasound procedures by implementing standardized protocols and training for sonographers. Second, the efficacy of this method of assessment is not evaluated in different surgeries and must be studied in the future. Third, this was a single-center study, and future multicenter studies may confirm our findings in larger patient populations. Fourth, the exclusion criteria included body mass index exceeding 35 kg/m$^2$ in our trial, but obesity or physical changes may interfere with the accuracy of diaphragmatic and lung ultrasound, affecting the reliability of U-scores in such populations. Therefore, subgroup analysis should be carried out in the future to expand on the findings.

## CONCLUSIONS

With validation, the U-score using both LUS and DUS may provide a option for estimating pulmonary function. This approach facilitates the assessment of pulmonary function in patients who may be unable to reliably participate in pulmonary function testing. This model can be easily integrated into existing clinical workflows, as ultrasound data can be collected during routine examinations. This allows patients at high risk of postoperative complications to be identified early, and interventions can be initiated in a timely manner.

## ACKNOWLEDGEMENTS

The authors thank Prof. Xiaoping-Zhu for his valuable assistance.

### Funding

This work was supported by Health Commission of Jiangxi Province (202210389). The funders had no role in study design, data collection and analysis, decision to publish, or preparation of the manuscript.

## Grant Disclosures

The following grant information was disclosed by the authors:
Health Commission of Jiangxi Province: 202210389.

## Competing Interests

The authors declare that they have no conflicts of interest.

## Author Contributions

- Tianyuan Li conceived and designed the experiments, performed the experiments, prepared figures and/or tables, authored or reviewed drafts of the article, and approved the final draft.
- Xiong-zhi Wu conceived and designed the experiments, performed the experiments, analyzed the data, prepared figures and/or tables, authored or reviewed drafts of the article, and approved the final draft.
- Dingde Long conceived and designed the experiments, performed the experiments, analyzed the data, prepared figures and/or tables, authored or reviewed drafts of the article, and approved the final draft.
- Huan Fu conceived and designed the experiments, performed the experiments, analyzed the data, prepared figures and/or tables, authored or reviewed drafts of the article, and approved the final draft.
- Suping Guo conceived and designed the experiments, performed the experiments, prepared figures and/or tables, authored or reviewed drafts of the article, and approved the final draft.
- Fen Liu conceived and designed the experiments, authored or reviewed drafts of the article, and approved the final draft.

## Human Ethics

The following information was supplied relating to ethical approvals (*i.e.*, approving body and any reference numbers):

This study was approved by the Ethics Committee of First Affiliated Hospital of Nanchang university (NO. 2021-8-001).

## Clinical Trial Ethics

The following information was supplied relating to ethical approvals (*i.e.*, approving body and any reference numbers):

The trial was registered at the Chinese Clinical Trial Registry (No: ChiCTR2100048032; URL: www.chictr.org.cn).

## Data Availability

The raw measurements are available in the Supplemental File.

## Supplemental Information

Supplemental information for this article can be found online at http://dx.doi.org/10.7717/peerj.18677#supplemental-information.

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
