# Peer review of "Ultrasonographic modeling of lung and diaphragm mechanics: clinical trial of a novel non-invasive method to evaluate pre-operative pulmonary function"

_PeerJ, doi:10.7717/peerj.18677_

## Round 0.1 · original submission · Major Revisions

Dear Authors,

Please make revisions to your manuscript according to reviewers suggestions or write a detailed rebuttal on a point-by-point basis.

Reviewer 1 ·

Basic reporting

The title "Ultrasonographic modeling of the lungs and diaphragm: A novel technique to assess pulmonary function in preoperative patients" is normally appropriate, but its effectiveness depends at the desires of the paper and the audience. Here are some wonderful aspects and suggestions for development:

Positive Aspects:

Clarity: The title virtually communicates that the study entails ultrasonographic modeling of the lungs and diaphragm.
Specificity: It immediately indicates that the examine makes a speciality of preoperative sufferers, that's vital for readers looking for paintings on this location.
Novelty: The word "novel technique" signals that the study presents some thing new, that may appeal to a scientific target audience.
Potential Suggestions for Improvement:
Specificity: If viable, upload more element approximately what exactly is novel in the approach or the capability results.
For example: "Ultrasonographic modeling of lung and diaphragm mechanics: A novel non-invasive method to evaluate pulmonary characteristic in preoperative sufferers"
Precision in Terminology: If "modeling" is the precise time period to explain the method, keep it. However, in case you're assessing feature rather than actually modeling, recall using phrases like "evaluation" or "assessment."

Overall, the title is applicable and informative.


Clarity and syntax:
In many places, the sentences are slightly inconsistent or difficult, which can hinder reading. Simplifying and reducing the sentences will make it easier to follow the text.

For example:
Instead of: "Pulmonary function testing (PFT) is an integral part of the intraoperative patient evaluation and clinically assesses functional impairment."
You could write, "Pulmonary function testing (PFT) is an important part of surgical evaluation of patients, and helps to monitor impaired function."
Synonyms:
Be consistent in the use of certain terms, such as "pulmonary dysfunction" and "pulmonary damage." choose one word and use it throughout the text for clarity and coherence.

Exactly the objectives of the survey:
At present, the objectives of the study are described in some detail. Consider clarifying exactly what "objective standard development" means and how the course plans to achieve this. It may be helpful to explain how a mathematical model would contribute to objective pulmonary function assessment.

Links to previous studies:
You mention several studies highlighting the usefulness of LUS in the diagnosis of respiratory diseases. However, it would be useful to link these studies to your own, especially by highlighting what your study’s new contributions are and how they fill gaps in existing knowledge.

Implicit hypothesis formulation:
In the last part of the introduction, you mention that no correlation has yet been established between LUS and DUS in combination with PFT. But it is unclear whether your hypothesis is that you would have revealed this relationship. Clear formulation of the research question or hypothesis can reinforce explanation identification.

Experimental design

Here are the specific issues I have found with your content and methods department.

1. Redundancy in curriculum
The curriculum has been described several times in an iterative manner. For example, terms such as "prospective cross-sectional study" and "prospective observational study" are used interchangeably. You have to define exactly one and stick to it throughout the lesson.
Tip: Make the design clear first and avoid repetition.

2. Lack of specificity in ultrasound techniques
Descriptions of LUS and DUS techniques may be more extensive, especially for readers unfamiliar with ultrasound. For example, it would be useful to elaborate on scoring systems or criteria when referring to "pulmonary ventilation" or "pulmonary circulation."
Recommendations: Provide detailed steps on how the ultrasound scan was performed and scored for repeatability.

3. Unclear about methods Objectives of the study
The purpose of the study is clear in the introduction but less clear in the materials and methods section. When introducing the methods, it is helpful to briefly remind readers that you are trying to find a relationship between LUS/DUS and PFT.
Suggestion: Re-emphasize the purpose of the lesson when describing the methods, so that readers understand the logic of the method.

4. The Statistical Methods section is repetitive
The section on statistical methods is unnecessarily repetitive. You’re explaining the same math test over and over again, which can confuse readers and make it more difficult to follow the story.
Recommendation: Combine descriptions of statistical tests into one coherent paragraph, indicating which tests were used for continuous classification, and avoiding repetition of terminology.

5. Misinterpreted ultrasound indices
The section on diaphragm ultrasound (DUS) indices, especially diaphragm thickness fraction (DTF) and excursion (DE), would benefit from a more detailed description of the accuracy with which these values ​​are measured and their implications for diaphragm function.
Recommendations: Provide a brief explanation of the physiological significance of indices such as DTF and DE as why they are important for lung function assessment.

6. Improve word accuracy
Some words are used inconsistently. For example, "pulmonary dysfunction" and "pulmonary dysfunction" seem to be used interchangeably, but using the same term throughout would lead to confusion
Tip: Match terminology to ensure clarity and consistency.

7. Submission of results
Although part of the results section, it bears some resemblance to Materials and Methods. Feedback is straightforward, but the transition from processes to outcomes can be easy. You introduce a lot of data into this section (e.g., LUS scores, D-DE, Q-DFT, etc.) without first setting clear expectations about what you are measuring and why.
Tip: In content strategies, it would be helpful to know the key metrics to analyze (LUS, D-DE, DTF) so that readers can anticipate these results when they view later.

8. Ultrasonographic specimen score
The ultrasonographic modeling score (U-score) has been described near the end, but the method is not well defined. Details on how these scores were developed, the variables they included, and how they were validated should be described in detail in the Materials and Methods section.
Suggestions: Explain how the U-scores were derived from and with the LUS and DUS measurements

Validity of the findings

1. Reinforce the first paragraph
The first paragraph provides strong information but can be more directly relevant to your study. Nowadays, it reads more like a general literary review. It will help to highlight how your study meets the limitations of PFT in specific populations.
Recommendation: Reframe the paragraph to emphasize how your study addresses this gap, for example: “Although preoperative PFT is important for predicting postoperative complications, but the effectiveness is low in a number of populations.”

2. Linking results
The conversation shifts somewhat abruptly from general research to results. Try to relate the introductory part of the discussion more closely to the main findings of your research, especially with regard to the substitution of ultrasound for PFT.
Suggestion: Include a generalizing sentence with your findings, such as: "In this study, we developed an ultrasound model to assess lung function during surgery and identify major risk factors." the lungs don't work properly."

3. Clear interpretation of the results
In discussing the findings (e.g. the relationship between D-DTF and FEV1/FVC) it would be helpful to provide more context as to why these results are important. This section of the discussion lists outcomes without fully explaining their clinical implications.
Recommendations: Expand the clinical relevance of these results. For example: "The correlation between D-DTF and FEV1/FVC suggests that ultrasound measurements can provide an accurate pulmonary function assessment in patients who cannot cooperate with PFT. This is especially valuable for patients who cannot." don't go anywhere or they're confused."

4. A clear description of the U-score
The U-score is an important part of your study, but it needs to be fully explained in the discussion. Although you talk about its benefits, it’s not clear how it beats individual LUS and DUS results.
Suggestion: Explain why the U-score is more accurate, for example: "The U-score combines LUS and DUS results to obtain a comprehensive assessment of lung function. This overcomes one of the limitations of these tools." overcome the will to use it, as we see it." with high specificity and sensitivity for pulmonary dysfunction." "

5. Limitations of the study
The boundaries are well stated, but could have been more detailed. In particular, it would be useful to include a discussion of how the reliability of ultrasound techniques can be improved, given their reliance on operators.
Recommendations: Include suggestions for future improvement, such as: "One way to improve the accuracy of ultrasound procedures is to implement standardized protocols and training for sonographers." Future multicenter studies may confirm our findings in larger patient populations."

6. Relationship to clinical practice
You thoroughly explain the medical implications of the model, but it would be helpful to add more on how to apply the model to everyday practice. The discussion should include specific examples of how clinicians can use the U-score in decision-making.
Recommendations: "This model can be easily integrated into existing clinical workflows, as ultrasound data can be collected during routine examinations. This allows patients at high risk of postoperative complications to be identified." early, and intervenes in a timely manner."

Additional comments

Technical validation and generalization:
Although the U-score model offers an alternative, further validation is required in a broader patient population to ensure its effectiveness in an experimental group as noted in pulmonary studies of ultrasound (LUS), its accuracy and applicability may vary among clinical settings Trials are needed in order to accommodate large sample sizes at multiple sites

Study Limitations and Patient Variability: Your criticisms regarding patient selection and lack of detailed description of variability in pulmonary disease іs well based pulmonary artery) may significantly affect the results. Therefore, it is important to investigate the effectiveness of the U-score in different diseases, as suggested by studies focusing on diaphragmatic and pulmonary ultrasound in conditions such as IPF and COPD.

Effect of body weight and anatomy:
Ultrasound measurements are highly dependent on patient anatomy and body weight. For example, obesity оr physical changes may interfere with the accuracy of diaphragmatic and lung ultrasound, affecting the reliability of U-scores in such populations Studies confirmed this limitation, and it was enforced note that physical features play an important role in ultrasound imaging, especially in moderate to severe or obese patients


Limitations in acute medical conditions:
While the U-score model can be useful in prescribing medical diagnoses, its complications in acute care or emergency situations can be a challenge In emergency situations, where rapid decision-making is important, the time it takes to calculate detailed scores, . That may reduce its usefulness Research shows that although LUS is useful for detecting conditions such as respiratory failure, its usefulness in an emergency situation may be limited


Consequences of exaggerating observations:
Although the U-score model shows promising associations with established lung function indicators (such as FEV1/FVC) are still in the early stages of development, similar findings have been found in LUS and diaphragm ultrasound in highlights that these mechanisms are not yet ready to replace conventional pulmonary function. Because extensive testing and modification is needed.

Reviewer 2 ·

Basic reporting

The authors of this paper have examined the role of Lung and diaphragm US in pre-operative as a substitute for pulmonary function assessment. Overall, the manuscript overall is fairly well written although there are some uncommon spelling errors as well as grammatical and stylistic issues:
- Lines 157 and Figure 1 section F: “score” instead of “socre”
- Figure 2 title: “Trial” instead of “Trail”
- Lines 130-137: Statistical analysis has been repeated in lines 134-137.
- Figure titles are mislabeled: Figure 1 in the legend is labelled as Figure 3 in the image, Figure 3 in the legend is labelled as Figure 4 in the image, and Figure 4 legend is labelled as figure 2 in the image
- Table 2: Age is reported as P < 0.01 by the ** but only a single star should be used as the P value is 0.019

The introduction describes the role of LUS as an alternative to pulmonary function testing as it does not require the patient to follow a number of complex instructions. However, the authors need to explicitly state the reason that DUS is an important for assessing lung function. They cite literature in lines 72-74 that suggests as association with lung function in specific disease states but do not discuss the limitation of LUS and how DUS can fill that gap. Perhaps stating in the opening line of each paragraph that DUS can assesses respiratory muscle function and LUS examines lung fields and the pleura.

The authors have provided their statistical output to the journal per PeerJ policy.

Experimental design

This study is the first to examine the correlation between LUS/DUS and PFT. The experimental design is a prospective cross-sectional study with blinding for the US performance. The authors apply appropriate correlation studies and then determine the US measurements with strongest correlation to PFT for ROC analysis and to develop a statistical model to predict pulmonary function. There are some questions or comments listed below that should be addressed.

The authors, however, need to clarify some details regarding their US measurements. They have not reported the use of a linear probe for lung ultrasound (Lines 107-113) and diaphragm thickness measurements (Lines 119-123). Typically, DT is measured with a linear probe to provide a non-curved image of the diaphragm for measurements. The authors have also used B-mode to measure diaphragm thickness but the typical standard is to use M-mode to measure diaphragm thickness as the same cross section of the diaphragm is compared in M-mode.

In the methods section (Line 96-98), the authors also describe the study procedure to complete the US assessments on the same day as the PFT, which requires exertion for FVC and FEV1. But the LUS and DUS are separated by 5 hours. This reasoning is not explained.

Have the authors considered correlating LUS and DUS with clinical outcomes in addition to PFT to predict the likelihood of post-operative complications? In this cross section, how predictive was PFT. The authors mention 2 cases of normal PFT have post op complications. Did all abnormal PFT have post-operative complications?

Validity of the findings

Results are valid as reported based on the statistical analysis but there are some questions mentioned below.

Is the logistic regression univariate or bivariate, as regression typically is examining a relationship between x and y?

The authors, however, may wish to describe the types of surgery that were planned for this cross-section.

The authors describe a maximum BMI to omit patients with excess adipose tissue that would limit the ability to acquire data. Were there exclusions for low BMI?

The authors describe an interesting role of sex in DE (Lines 199-205). They state that sex is correlated with DE but was not an independent risk factor in the multi-variable model. Can the authors explain this disparity as they have provided biologic plausibility for these differences but their data does not show this difference?

Additional comments

Line 191 - The authors state that they have developed ultrasound models but I think the authors should state that they have devised statistical models using US measurement to predict PFT.

The authors state that a limitation of their study is the lack of uniform criteria (Lines 237-241). There is a new consensus statement (Demi 2023) for LUS as well as an older statement (Volpicelli 2012) that addresses the role LUS including regions of the chest to scan although the focus on the older statement was emergency and critical care:

• Demi L, Wolfram F, Klersy C, De Silvestri A, Ferretti VV, Muller M, Miller D, Feletti F, Wełnicki M, Buda N, Skoczylas A, Pomiecko A, Damjanovic D, Olszewski R, Kirkpatrick AW, Breitkreutz R, Mathis G, Soldati G, Smargiassi A, Inchingolo R, Perrone T. New International Guidelines and Consensus on the Use of Lung Ultrasound. J Ultrasound Med. 2023 Feb;42(2):309-344. doi: 10.1002/jum.16088.

• Volpicelli, G., Elbarbary, M., Blaivas, M. et al. International evidence-based recommendations for point-of-care lung ultrasound. Intensive Care Med 38, 577–591 (2012). https://doi.org/10.1007/s00134-012-2513-4

The authors mention a second limitation surgeries (Line 237-241) is that LUS/DUS is not evaluated for different. These details regarding the type of surgery are missing in the results.

The authors do not mention another limitation is that those who completed the US measurement are not blinded to the clinical history, which can introduce bias. They only explicitly mention that the individual obtaining the US images is blind to the clinical history and PFT.

In the discussion, the authors report data not mentioned in the results section (line 220-230). This data is intended to illustrate some limitations of PFT. The authors however did not clarify if any patients with abnormal LUS/DUS had a normal post-operative course.

---

## Round 0.2 · Minor Revisions

Please make minor revisions suggested by the reviewer and provide a rebuttal in detail on a point-by-point basis.

Reviewer 2 ·

Basic reporting

The authors of this paper have addressed most of the reviewer comments. The manuscript overall has some minor spelling, grammatical and stylistic errors that can be addressed:
- Title: I suggest changing “pre-operative suffers” to “A novel non-invasive method to evaluate pre-operative pulmonary function”
- Key words: I suggest adding pre-operative assessment
- Abstract (line 22-23): I suggest changing the first sentence to the following “Pre-operative PFT plays a key role in predicting post operative complications or functional impairment.”
- Abstract (line 26): I suggest to remove “Patients’” from patients’ dyspnea and write ”Dyspnea is mainly caused by…”
- Abstract (line 27-29): You can combine the following sentences to improve the flow: “As the diaphragm is a primary respiratory muscle, combining lung with diaphragm ultrasound may enable a more comprehensive assessement of pulmonary function.”
- Introduction (line 54-56): I suggest changing to: “Pre-operative PFT parameters including forced expired volume in the first second as a fraction of forced vital capacity are important predictors of post-operative complications.”
- Introduction (line 65): I suggest changing to “examinations across a variety of pulmonary disease states and clinical settings”
- Methods (line 103): please change “patient decline” to “patients not providing consent”
- Methods (line 108): please change “clinical datas” to “clinical data”
- Methods (line 160-161): The authors may need to more clearly state: The left diaphragm was not assessed due to difficulty with image acquisition related to the gastric bubble and intestinal gas”
- Methods (line 162): Please change “Probe over the opposition zone” to “probe over the zone of apposition”
- Methods (line 181-183): Please review as this is an incomplete sentence with incorrect capitalization.
- Methods (line 183-186): This is a run on sentence. Please separate with semi-colon or period. “…and area under the curve (AUC); receiver operating characteristic (ROC)…”
- Results (line 188): This is a run on sentence. Change to “Of all 208 patients included in the study, 35 patients…”
- Results (line 190-191): I suggest clarifying to “174 patients exhibited normal pulmonary function testing”
- Discussion (line 222-224): I suggest changing to “Among elderly patients with increasingly complex health conditions, reduced mobility, inadequate communication skills, cognitive impairment, ventilator dependency or poor cooperation may affect the ability to complete PFT.”
- Discussion (line 226): what population is the effectiveness of PFT low? If it is elderly, patients, then it would be relevant but if it is a specific age group, sex or ethnicity then this sentence may not apply.
- Discussion (line 234): Change “a minimum cooperation” to “minimal cooperation”
- Discussion (line 238): Remove the extra period
- Discussion (line 304): I suggest changing to “identified early, and interventions can be initiated in a timely manner”
- Figure 1: “208 patients analyzed” instead of “208 patients analysis” and “174 normal pulmonary function testing” instead of “174 abnormal pulmonary function” and “34 abnormal pulmonary function testing”
- Figure 4: “U-score” instead of “U-socre”
- Table 2: Age is reported as P < 0.01 by the ** but only a single star should be used as the P value is 0.019

Experimental design

The authors employ logistic regression to determine the association between LUS/DUS and abnormal pulmonary function testing. I suggest that the author change multivariate to multivariable as multivariate suggest that more that one outcome is measured such as repeat measures.

Validity of the findings

The authors describe the role of sex in DE (Lines 199-205) but report that no differences were found in this study. The authors state that the lack of difference in DE by sex is due to no difference in sex between group. If sex is an independent risk factor, then having both groups balanced should reveal this difference. They can consider omitting the biologic explanations as they seem to suggest that sex should be an independent predictor but it may be an area that requires further study.

Additional comments

The authors mention 2 cases of normal PFT have post op complications. I suggest adding the details of there LUS/DUS and post-operative course to the results section. The explanation of the difference in US and PFT can be in the discussion.

---

## Round 0.3 · accepted · Accept

Dear Authors, your paper is now acceptable for publication in its current form.